# Biomarkers and Corresponding Biosensors for Childhood Cancer Diagnostics

**DOI:** 10.3390/s23031482

**Published:** 2023-01-28

**Authors:** Azadeh Gharehzadehshirazi, Mashaalah Zarejousheghani, Sedigheh Falahi, Yvonne Joseph, Parvaneh Rahimi

**Affiliations:** 1Institute of Electronic and Sensor Materials, Faculty of Materials Science and Materials Technology, Technische Universität Bergakademie Freiberg, 09599 Freiberg, Germany; 2Freiberg Center for Water Research—ZeWaF, Technische Universität Bergakademie Freiberg, 09599 Freiberg, Germany

**Keywords:** cancer, childhood, biomarker, biosensor, early-stage diagnosis

## Abstract

Although tremendous progress has been made in treating childhood cancer, it is still one of the leading causes of death in children worldwide. Because cancer symptoms overlap with those of other diseases, it is difficult to predict a tumor early enough, which causes cancers in children to be more aggressive and progress more rapidly than in adults. Therefore, early and accurate detection methods are urgently needed to effectively treat children with cancer therapy. Identification and detection of cancer biomarkers serve as non-invasive tools for early cancer screening, prevention, and treatment. Biosensors have emerged as a potential technology for rapid, sensitive, and cost-effective biomarker detection and monitoring. In this review, we provide an overview of important biomarkers for several common childhood cancers. Accordingly, we have enumerated the developed biosensors for early detection of pediatric cancer or related biomarkers. This review offers a restructured platform for ongoing research in pediatric cancer diagnostics that can contribute to the development of rapid biosensing techniques for early-stage diagnosis, monitoring, and treatment of children with cancer and reduce the mortality rate.

## 1. Introduction

A cancer diagnosis is always distressing, particularly when the patient is a child, since having cancer causes many changes in a child ‘s life. Cancer is a major cause of death in children around the world. It is estimated that each year, around 400,000 children and adolescents from 0–19 years old are diagnosed with cancer worldwide [1]. Cancers are caused by a change in one or more genes in single cells, which then grow into a tumor. Contrary to cancers in adults, the majority of cases in children are idiopathic. However, about 10% of cases are due to germline mutation, which is almost always not inherited, i.e., family history cannot predict the presence of cancers in these patients [2]. A number of factors can impact the outcome of a tumor, including the biological characteristics and pathway of a neoplasm’s carcinogenesis, modes of presentation, time to diagnosis, and treatment options. Sometimes, the initial symptoms may be confused with other illnesses’ conditions, particularly in pediatrics cases, which makes it difficult to suspect a tumor [1,3]. The chance of a child surviving a cancer highly depends on the region where the child lives, as most cases of cancers in children need advanced technologies and specialists for diagnosis, effective therapies, and support, which are usually unaffordable and unavailable in low- and middle-income countries (LMICs) [4]. Generally, in high-income countries, more than 80% of children diagnosed with cancer are cured, whereas the cure rate in LMICs is less than 30% [3,4].

The types of cancers in children are entirely different from the ones affecting adults. In general, the most common children’s cancer types by proportion of overall child cancer rates are leukemias, followed by central nervous system (CNS) tumors and lymphomas. However, the type of cancer can vary to some extent by age [5]. Screening and diagnostic tests for the detection of cancers have always been challenging in both adults and children. Traditional tumor biomarkers that are found in blood, urine, cerebrospinal fluid (CSF), or even body tissues and are raised in connection with cancer have been used in oncology for a long time. Furthermore, different imaging modalities such as X-ray (plain film and computed tomography [CT]), ultrasound (US), magnetic resonance imaging (MRI), single-photon emission computed tomography (SPECT), positron emission tomography (PET), and optical imaging have been available in areas of cancer detection and follow-up. However, these methods are expensive, require highly qualified personnel, and may not be accurate, sensitive, or specific enough to detect a tumor. On the other hand, overdiagnosis by existing methods is still a major concern, since there is a need to avoid exposing a patient to invasive treatments. Additionally, biopsy as a definite diagnostic test is an invasive technique that sometimes needs short admission. Furthermore, the earlier and more precise diagnosis of cancer maintains a higher chance of cure. In this regard, rapid, sensitive, and specific detection of cancer biomarkers using biosensors has great clinical significance for the diagnosis of various cancers [6,7].

## 2. Potential Biomarkers for Childhood Cancers

Advancements in biochemistry and analytical instruments have played an important role in the identification of appropriate biomarkers and the development of corresponding biosensors, leading to significant improvements in treatment of patients in all age groups [8]. In accordance with the National Cancer Institute, a biomarker is “a biological molecule found in blood, other body fluids, or tissues that is a sign of a normal or abnormal process, or of a condition or disease “. Cancer biomarkers are biological factors that are identified to develop decisive, cost-effective, and non-invasive agents for early detection and monitoring of cancers. These biomarkers consist of a broad range of molecules, such as DNA, RNA, enzymes, metabolites, transcription factors, and cell surface receptors, that can be measured objectively by various techniques [9,10]. 

Due to the wide range of pediatric cancers and constant advances in molecular diagnostic techniques, in this review, we have focused on the most common cancer group in children. 

### 2.1. Leukemias

Leukemias, which are defined as abnormal proliferations of immature white blood cells (WBC), are kinds of hematologic malignancies that do not form solid neoplasms. Leukemias are the most common type of cancer in children, accounting for 28% of all cancers in children [11,12]. Leukemias are originated either from myeloid or lymphoid lineages, and they are categorized naturally as acute (non-mature cells) and chronic (more mature cells) [13]. More than 80% of leukemias in children are acute lymphoblastic leukemias (ALL) and the rest are mostly acute myeloid leukemias (AML). Chronic leukemias are very rare in children; however, most cases are chronic myelogenous leukemia (CML) [14]. Symptoms of anemia, thrombocytopenia, neutropenia, and persistent bone pain are typical leukemia presentations due to clonal proliferation of blasts in the bone marrow. Fever, hepatosplenomegaly, and CNS involvement are other manifestations of children with leukemia. Many of these symptoms overlap with other common childhood illnesses and make the diagnosis of leukemia complicated. In addition, acute leukemias can grow rapidly, so they need to be diagnosed and treated as soon as possible [15]. 

Several biomarkers have been identified and reviewed for screening, diagnosis, prognosis, and treatment monitoring in childhood leukemia. Among them, a cluster of differentiation (CD) markers, a series of membrane proteins, are widely used for classifying WBCs and also diagnosing blast cells from normal WBCs. CD19, CD20, CD22, CD 24, and CD79a are important markers for diagnosis and differentiating Burkitt type ALL (B-ALL) from other types of leukemias. In pro-B ALL (an unfavorable subset of ALL), lymphoblasts are positive for CD19, CD22, CD34, cytoplasmic (cy) CD79a, and TdT. Lymphoblasts in pre-B ALL are positive for CD10, CD19, CD22, CD34, TdT, cy CD79a, and Cy mµ. Thymic ALL (T-ALL) are positive for CD1a, cy CD3, CD5, CD7 and TdT [16,17]. Furthermore, there are specific genetic abnormalities of leukemic cells that play important roles as biomarkers in prognosis and therapy of leukemias. A number of these chromosomal abnormalities that act as potential markers include t(9;22), t(12;21), t(8;21), t(15;17), BCR-ABL, TEL-AML1, AML1-ETO, and PML-RARA [18,19]. Proteomic biomarkers that are other potentially interesting biomarkers for early diagnosis of ALL were reported by Shi et al. [20]. They identified platelet factor 4 (PF4), a kind of connective-tissue-activating peptide (CTAP III), and fragments of the complement component 3a (C3a) as proteomic biomarkers in the sera of ALL patients [20]. In addition, several miRNAs have been identified in patients with ALL, which have diagnostic and prognostic values. For instance, miRNA-328 and miRNA-200 are suggested as two novel biomarkers for diagnosis and prognosis, as well as miRNA-324-3p, miRNA-508-5p and miRNA-128 as potential diagnostic biomarkers of childhood ALL. [21,22,23]. Among other biomarkers related to leukemia, human leukemic lymphoblasts (CCRF-CEM) are cellular biomarkers that can be detected in the peripheral blood of children with acute leukemia [24].

### 2.2. CNS Tumors

Brain and spinal tumors are the second-most-common group of cancers and the leading cause of morbidity and mortality in children, representing about 25% of childhood cancers [25]. CNS tumors are normally classified on the basis of histologic features of the tumor and its probable site of origin in the brain. Nonetheless, the most common types of brain and spinal cord tumors in children are medulloblastoma (MB) and glioma [26]. Brain tumors in children are mostly infratentorial and cause acute manifestations related to a blockage in the flow of the CSF (headache, nausea, vomiting, irritability, abnormal breathing, etc.). However, in the long term, they may present with macrocephaly, delayed psychomotor development, loss of appetite, etc. [27]. Neuroimaging is the primary diagnostic technique for any kind of brain tumor; nonetheless, recently identified potential diagnostic, prognostic, and predictive biomarkers have become key tools for the management of these tumors [28,29]. CSF is considered as the most productive source for biomarkers in childhood CNS tumors because of its close vicinity to the tumor mass [30]. In a study, six proteins in CSF samples of patients with CNS tumors were identified as prospective biomarkers for metastasis spread, including type 1 collagen, insulin-like growth factor binding protein 4, procollagen C-endopeptidase enhancer 1, glial-cell-line-derived neurotrophic factor receptor α2, inter-alpha-trypsin inhibitor heavy chain 4, and neural proliferation and differentiation control protein-1 [31]. Recently, Bruschi et al. introduced six promising biomarkers from CSF of children with different brain tumors. Among these biomarkers, TATA-binding protein-associated factor 15 and S100 protein B were able to differentiate between control and tumor cases [32]. TMSB4X, a cytoskeletal protein inhibiting actin polymerization involved in tumorigenesis, and CD109, a glycosylphosphatidylinositol-anchored cell surface antigen expressed by T-cells and endothelial cells, are biomarkers found in CSF of patients with pilocytic astrocytoma (PA), a low-grade cerebellum glioma, that can discriminate PA from all other brain tumors [33,34,35]. Another instance is tripartite motif-containing protein 33 (TRIM33), which is a transcriptional corepressor suppressor of brain tumor development that can distinguish PA from the more aggressive diffuse gliomas [36,37]. Recently, Bookland et al. [38] introduced miRNA-21, miRNA-15b, miRNA-23a, and miRNA-146b as four miRNAs that can predict the presence, tumor nodule size, and response to therapy of PA with a high sensitivity and specificity. Heat shock proteins (HSPs), which belong to a superfamily of chaperones, along with 14.3.3 as an inhibitor of apoptosis are overexpressed in MB and can discriminate between MB and all other brain tumors [39,40,41]. Circ NFIX RNA is another biomarker that is considerably upregulated in glioma cancer cells and has attracted scientists’ attention, since Circ NFIX could regulate signaling pathways leading to human glioma cancer progression [42]. Among the biomarkers, glioblastoma-derived specific exosomes, which release directly from tumor cells in brain and cross the blood–brain barrier, hold great potential for early diagnosis of glioblastoma (GBM, a type of glioma) [43].

### 2.3. Neuroblastoma

Neuroblastoma (NB) is the most common cancer diagnosed in the first year of infancy [44]. This embryonal malignancy is derived from primordial neuronal crest cells and is mostly located in adrenal medulla and along the sympathetic nervous system [45]. The NB’s symptoms depend on the anatomic location of the tumor and the stages of the disease. Most patients have presentations related to a mass in the belly, such as abdominal distention, constipation, and high blood pressure. In cases arising from paravertebral sympathetic ganglia, the spinal cord is usually invaded and patients may have manifestations such as pain, motor sensory deficit, or Horner syndrome (drooping eyelid, small pupil, and lack of sweating on one side of face) [46,47,48,49]. There are multiple imaging techniques and tests for diagnosis of neuroblastoma. Ultrasonography is usually the first modality in a child with a suspicious abdominal mass, followed by CT or MRI for further evaluation of the tumor extension and stage [50]. Furthermore, nuclear medicine imaging techniques such as meta-iodobenzylguanidine (MIBG) and fluorodeoxyglucose (FDG) imaging are often used to define tumor status and detect occult disease and probable metastasis [51]. Along with imaging, there are several biomarkers in neuroblastoma that are valuable in the process of tumor accessing. They can be divided into two main groups: 1) Amplification of genetic and molecular biomarkers such as MYCN (a protein-coding gene), ALK (anaplastic lymphoma receptor tyrosine kinase gene) amplification and mutations, chromosomal segments loss or gain, and dysregulated proteins such as PIM kinase, Far-Upstream Element-Binding Protein 1 (FUBP1), Ubiquitin C-Terminal Hydrolase L1 (UCHL1), and Tropomodulin1 (TMOD). 2) Circulating biomarkers in serum such as neuron-specific enolase (NSE), circulating tumor cells (CTCs), and miRNAs (such as overexpression of miR-124-3p, miR-9-3p, miR218-5p, miR490-5p, or miR1538), as well as circulating biomarkers in urine, including catecholamines and their derivatives such as 3-Methoxytyramine (3-MT), homovanillic acid (HVA), vanillylmandelic acid (VMA), normetanephrine (NMN), and vanillactic acid (VLA) [52,53,54].

### 2.4. Wilms Tumor

Wilms tumor (WT) or nephroblastoma is the most common kind of primary kidney tumor and the third-most-common solid tumor in children, accounting for about 5% of all childhood cancers [55,56]. Wilms tumor may be associated with different congenital anomalies and syndromes such as WAGR syndrome (a rare genetic disorder and acronym for Wilms tumor, Aniridia, Genitourinary anomalies, and mental Retardation), Beckwith–Wiedemann syndrome (macrosomia, macroglossia, abdominal wall defects, Wilms tumor, hypoglycemia in the newborn period, and unusual ear creases or pits) and Denys–Drash syndrome (abnormal kidney function, disorders of sexual development in affected males and Wilms tumor) [57,58]. The most usual feature of WT is a large and painless abdominal mass in a clinically healthy child [59]. However, about 20% of patients with WT also show symptoms such as abdominal pain, constipation, urinary tract infections, blood in the urine, high blood pressure, fever, and weight loss [60,61].

A child with suspected WT is typically examined with imaging techniques in the first stage. Ultrasound is normally the first choice, which is then followed by further imaging such as CT or MRI in order to define the stage of the tumor [62,63].

There are a number of biomarkers that have been most identified for their role in WT characteristics and histology. Among them, loss of heterozygosity (LOH) for both chromosomes 1p and 16q, which is closely associated with tumor recurrence; mutation in B7-H1 (programmed death-ligand 1 = PD-L1) as a biomarker of the immune system, which is related to an increased risk of tumor recurrence; and mutation in P53, which is involved in an increased risk of recurrence, are promising biomarkers in identifying WT [64,65,66]. Furthermore, Ludwig et al. [67] have recently identified miR-100-5p, miR-130b-3p, and miR-143-3p with diagnostic potentials for differentiation of WT from controls. They reported that among three recognized miRNAs, higher expression of miRs-100-5p and -130b-3p in the serum of patients with WT has the greatest potential for tumor diagnosis regardless of its subtype.

### 2.5. Lymphomas

Lymphomas originate in transformed cells of the lymphatic system (germinal-center B cells). There are two major groups of lymphomas, Hodgkin lymphomas (HL) and non-Hodgkin Lymphomas (NHL), and both types occur in children. However, HL is the most common lymphoma in children, accounting for approximately 6% of all children’s malignancies, and is most frequent in adolescents [68]. Lymphomas typically affect the whole body. Most patients with lymphoma present with non-tender lymph node swelling that may be followed by constitutional symptoms (fever, fatigue, loss of appetite, sweating, weight loss, or not gaining weight) [69]. Imaging plays a crucial role in the diagnosis, staging, and follow-up of patients with lymphoma. CT, MRI, and PET are conventional imaging modalities used in patients with lymphomas for further evaluations [70]. Most lymphomas are potentially curable when they are diagnosed and treated with a standard care method. However, recognizing different lymphoma subtypes can be challenging due to the overlapping features of various lymphoma groups. Thus, the identification of clinically valuable biomarkers may facilitate the diagnosis and classification of lymphoma, which in turn should lead to better stratification of patients and more efficient and less toxic treatment of children with lymphoma. In this regard, serum CD163 and serum thymus and activation-regulated chemokine (TARC) are proposed as striking biomarkers reflecting therapy response in patients with HL [71]. CCL17/TARC is another biomarker with the potential to facilitate primary care triage and chemotherapy monitoring strategies for classic Hodgkin lymphoma (cHL) [72]. Determining the levels of a combination of biomarkers such as Nuclear Factor Kappa B (NF-κB) and CD30 in pediatric HL patients would support gaining insight into disease progression during tumor treatment [56]. Additionally, four biological and inflammatory markers (stage IV, high platelet count, ferritin, and eosinophils) were investigated by Farruggia et al. to classify pediatric patients with HL into subtypes with various outcomes [73]. More recently, Yu et al. [74] identified S100 calcium-binding protein A8 (S100A8) and leucine-rich alpha-2-glycoprotein 1 (LRG1) as promising biomarkers for the diagnosis of pediatric NHL.

## 3. Biosensors for Detection of Biomarkers

Cancer biomarkers are commonly detected by conventional techniques, including enzyme-linked immunosorbent assay (ELISA), polymerase chain reaction (PCR), Western blotting, immunofluorescence, flow cytometry, and liquid chromatography–mass spectrometry [75,76,77,78,79]. Nevertheless, these assays are complex, time-consuming, require complicated and expensive instrumentation, and sometimes fail to detect a low concentration of biomarkers in the early stages of cancer. Therefore, the development of a reliable, rapid, and sensitive method for biomarker detection, especially in the early stages of cancer, is essential. In this regard, biosensors have shown attractive features compared to the techniques mentioned earlier and have improved health and quality of life. On the other hand, biosensors are promising candidates for the specific and simultaneous detection of biomarkers and the study of their associated reactions, because their components can be easily modified and improved.

A biosensor is an integrated receptor–transducer device capable of transforming a biological reaction into a measurable signal, which can be further amplified and analyzed (Figure 1) [80,81]. The biorecognition element has the most important function in the biosensor, as it defines the selectivity and sensitivity of the designed biosensor by specific binding to a target analyte. Therefore, selecting a suitable bioreceptor is the first step to improving the biosensors’ design and development [82,83]. At this point, various types of biosensors have been developed for the determination of cancer biomarkers. Many researchers have enhanced the sensitivity of biosensors by integrating nanomaterials into biosensing assays [84,85]. Novel chemical, physical, optical, and electrical properties of different nanomaterials and their large specific surface areas can significantly improve the ability of biosensors to detect biomarkers. In addition, it is proved that the detection of multiple biomarkers can remarkably advance the specificity of a biosensor for the diagnosis of early-stage cancer [85,86].

## 4. Developed Biosensors for Childhood Cancers in Clinical Practice

To date, numerous cancer biomarkers have been identified for various cancers in children and adults. Although a variety of biosensors have been developed for biomarker detection, only a limited number of them have been designed for childhood cancers. Here, the developed biosensors for biomarker detection related to childhood cancers are reviewed

### 4.1. Electrochemical Biosensors

Among the developed biosensing technologies for cancer biomarker detection, electrochemical biosensors have gained much attention due to their low cost, ease of use, suitability for miniaturization, fast response time, and high sensitivity with low detection limit. In particular, electrochemical nano-biosensors using various nanomaterials have been recently used for ultra-sensitive detection of biomarkers [87].

Mazloum Ardakani et al. [88] developed a set of disposable electrochemical nano-biosensors for early detection of acute lymphoblastic leukemia. Their biosensor package included a DNA sensor, an aptasensor that can detect BCR-ABL1 as a mutant gene, and carcinoembryonic antigen (CEA) as a cancer biomarker. To improve the sensitivity of their biosensors, they used a nanocomposite of carbon quantum dots and gold nanoparticles (Au NPs) for preparing both DNA and aptamer biosensors. With the proposed nano-biosensor package, they could detect BCR-ABL1 and CEA in a wide linear range with the detection limits of 1.5 and 0.95 pM, respectively. According to their research, a negative test result of DNA sensor was related to a person who did not have ALL. In contrast, a positive DNA test suggested that the patient had a mutant gene and was at risk of ALL. The aptamer test was performed to verify the positivity in case of a positive DNA test. A positive aptasensor test illustrated that the patient definitely had ALL. On the contrary, a negative aptasensor test would mean that the person had no ALL but was at risk of ALL because of having the mutant gene responsible for ALL.

Avelino et al. [89] developed a simple, rapid, and highly sensitive electrochemical DNA biosensor to diagnose BCR/ABL oncogene for both ALL and CML at attomolar concentrations. They used a composition of Au-NPs and polyaniline, an organic–inorganic hybrid material, with synergistic properties and robust biosensitive platforms, to attach DNA probes to the gold electrode. By applying complementary DNA (cDNA) samples from patients to the biosensor surface and hybridizing with immobilized DNA probes, the concentration of the BCR/ABL fusion gene could be determined at the lowest concentration compared with the other reported values.

Dinani et al. [23] fabricated an aptasensor based on Au NPs/Fe_3_O_4_/reduced graphene oxide (rGO) for the determination of miRNA-128 concentration as a diagnostic biomarker for ALL. They decorated the rGO films with Fe_3_O_4_ as a suitable substrate for easier deposition of AuNPs as well as rapid immobilization of aptamers on the surface of the electrode, which improved the electrochemical conductivity of the rGO sheets and the sensitivity of the fabricated biosensor. MiRNA-128 was then detected in low concentrations using label-free and methylene blue-labeled methods (Figure 2). Both label-free and labeled aptasensors showed high selectivity for miRNA-128. Their proposed platform could be integrated into wearable biosensors for real-time monitoring of individuals. Furthermore, compared to the previous studies, they could detect miRNA-128 at a very low concentration (fM), confirming the high applicability of the developed nano-biosensor for early detection of ALL.

Amouzadeh Tabrizi et al. [24] designed a flow injection aptamer–aptamer sandwich electrochemical biosensor for determination of CCRF-CEM cells in the presence of H_2_O_2_ as an electroactive component. They used a nanoplatform of poly (3,4-ethylenedioxythiophene) decorated with Au NPs for immobilizing the thiolated sgs8c aptamer and multiwall carbon nanotubes decorated with perylene tetracarboxylic acid and palladium NPs for catalytic labeling of aptamer. Then, the CCRF-CEM cells were sandwiched between them. The catalytically labeled aptamer bound to the CCRF-CEM cells catalyzes the electrocatalytic reduction of H_2_O_2_ and enhances the electrocatalytic reduction signal of H_2_O_2_ depending on the concentration of CCRF-CEM cells. With the proposed setup, they could also determine the CCRF-CEM cancer cells in human serum samples.

Recently, Rinaldi et al. [72] presented an electrochemical sandwich immunosensor as a point-of-care test for CCL17/TARC detection as a potential biomarker of cHL (Figure 3). The proposed sensor was able to distinguish patients with cHL from healthy volunteers as well as perform secondary care chemotherapy monitoring. In this regard, a gold electrode was first modified with a thiolated heterobifunctional crosslinker, Sulfo-LC-SPDP, to immobilize the specific capture antibody. Then, a biotinylated CCL17 secondary antibody conjugated with an enzyme was added to develop the final sandwich immunosensor for the determination of the CCL17/TARC level in the serum of a patient with cHL. The designed biosensor with lower and upper quantitation limits of 387 and 50,000 pg/mL towards CCL17/TARC showed a high dynamic range compared to the equivalent colorimetric ELISA platform, implying an essential step towards developing a rapid test for the staging and treatment of cHL.

A label-free electrochemical biosensor was designed based on Zr/metal-organic frameworks (Zr-MOFs) for early detection of GBM-derived exosomes in the blood of patients with GBM. These exosomes are marked by highly expressed human epidural growth factor receptor (EGFR) and EGFR variant (v) III mutations (EGFRvIII). Therefore, a peptide ligand was attached to the electrode surface as a sensing layer that could specifically bind to EGFR and EGFRvIII and capture GBM-derived exosomes. At the same time, encapsulated Zr-MOFs with methylene blue were combined with the GBM-derived exosomes through the interaction of phosphate groups of exosomes with Zr^4+^ and produced an electrochemical signal. The proposed method was able to quantitatively measure the concentration of GBM-derived exosomes, ranging from 9.5 × 10^3^ to 1.9 × 10^7^ particles/μL, with a detection limit of 7.83 × 10^3^ particles/μL. In addition, the prepared biosensing platform was used to analyze GBM-derived exosomes in human serum. It could distinguish GBM patients from healthy groups, demonstrating its feasibility for early diagnosis and monitoring of GBM therapy [43].

Moazzam et al. [90] fabricated an ultrasensitive and fast electrochemical biosensor for direct detection of PD-L1 in whole-blood samples using a dispersible electrode based on modified gold-coated magnetic nanoparticles (Figure 4). First, a labeled antibody conjugated with gold-coated magnetic nanoparticles (Ab1-Au@MNPs) was added to the PD-L1 solution in the blood. Then, the resulting magnetically separated PD-L1-Ab1-Au@MNPs were mixed with a second reporter antibody (biotinylated detection antibody) conjugated with a horseradish-peroxidase–streptavidin-labeled (HRP-Ab2) to build an immunosandwich structure of HRP-Ab2-(PD-L1)-Ab1-Au@MNPs. Then, the separated sandwiched HRP-Ab2-(PD-L1)-Ab1-Au@MNPs were collected on the gold macroelectrode surface (as working electrode) by applying a magnet, and PD-L1 detection was performed in ferrocenemethanol (Fc) solution as a redox mediator and in the presence of H_2_O_2_ as a substrate of HRP. The reported dispersible electrochemical sensor was able to detect PD-L1 with an ultralow detection limit of 15 attomolar, which is 2600 000 times lower than the detection limit obtained with commercially available ELISA kits.

### 4.2. Optical Biosensors

Optical biosensing technology is the second class of biosensors, after electrochemical, that has been extensively studied in cancer diagnostics and therapy. It can detect cancer biomarkers easily, directly, and in real time. Optical biosensors generally include colorimetric, fluorometric, chemiluminescence, and surface plasmon resonance (SPR)-based biosensors.

To develop a non-invasive liquid biopsy for early diagnosis of leukemia with high sensitivity and specificity, Huang et al. [91] proposed a dual-signal amplification fluorescent protocol to detect leukemia-derived nanosized exosomes. As shown in Figure 5, they first modified magnetic bead conjugates with anti-CD63 antibodies (MB-CD63) to entrap leukemia-cell-derived exosomes containing CD63 and nucleolin. Subsequently, a DNA primer with a nucleolin recognition aptamer (AS1411) was used to bind to the exosome, triggering a rolling circle amplification (RCA) reaction, to generate multiple repeated sequences for hybridization of fluorescent-labeled DNA (DNA-FAM), which was immobilized on AuNPs (GNP-DNA-FAM). Here, the fluorescence of FAM is quenched by AuNPs due to its specific optical effect. In the last step, they introduced nicking endonuclease (Nb·BbvCI) assisted target recycling, resulting in the release of FAM from the GNP-DNA-FAM conjugates, which transitioned from the quenching state to the emission state so that the fluorescence signals gradually increased. Using the suggested platform as a dual signal amplification strategy, they could detect exosomes at concentrations as low as 100 particles per μL.

Luo et al. [92] prepared a localized surface plasmon resonance (LSPR) biosensing platform for the specific detection of serum PD-L1 using excessively tilted fiber grating (ExTFG) coated with large-sized (∼160 nm) gold nanoshells. Anti-PD-L1 monoclonal antibodies were immobilized on an ExTFG-LSPR platform for label-free detection of PD-L1. The designed biosensor was able to specifically detect PD-L1 at a low concentration of 5 pg/mL in fetal bovine serum as a complex serum medium.

Recently, an LSPR biosensor chip was developed based on the immobilization of antibodies on the surface of Au nano-islands decorated with AgNPs (Ag@AuNIs) for the detection of monocarboxylate transporter 4 (MCT4) as a GBM progression biomarker [93]. The anti-MCT4 antibodies on the surface of Ag@AuNIs-modified chips had a remarkable ability to detect exosomes, resulting in the generation of a strong LSPR response. Therefore, the proposed biosensor sensitively and selectively detected the enhancement of MCT4 content in malignant hypoxic GBM cell-derived exosomes and the increased MCT4 content in exosomes from blood serum of GBM mice in a wide dynamic range, with a detection limit of 0.4 ng/mL, confirming its potential for early detection of GBM initiation and progression.

PD-L1 expressing extracellular vehicles (EVs) are of remarkable clinical relevance, since they have the potential to diagnosis cancer and to evaluate the patient’s response to anti-PD-L1/PD-L1 immunotherapy. In this context, Khan et al. [94] developed an aptamer-based chemiluminescence (CL) sensor for the detection of PD-L1@EVs using Fe_3_O_4_@TiO_2_ beads to capture EVs directly from undiluted serum without the need for additional ultracentrifugation and isolation. To improve specificity, a biotin-labeled PD-L1 aptamer was first added to bind to PD-L1@EVs. Streptavidin-modified alkaline phosphates (ALP) were then added to bind the aptamer via the strong biotin–streptavidin binding. The addition of CDP-Star, a chemiluminescent ALP substrate, triggers chemiluminescence proportional to the concentration of PD-L1@EVs. The prepared assay showed high sensitivity and specificity towards PD-L1@EVs in a wide linear range of 10^5^ to 10^8^ EVs/mL and a very low detection limit of 2.584 × 10^5^ EVs/mL.

Recently, electrochemiluminescence (ECL) (the combination of electrochemical methods and chemiluminescence) has gained popularity in sensor and biosensor applications due to the numerous advantages of ECL over chemiluminescence, such as controlled selectivity by changing the electrode potential, higher sensitivity, and time and position control of the light-emitting response. An ECL aptasensor for the detection of miRNA-16 in patients with leukemia based on polymerase-assisted signal amplification and aggregation of the illuminator was reported by Zhang et al. [95]. The assembled mercapto-modified capture DNA (CP) on Fe_3_O_4_@Au NPs was immobilized on the surface of a magneto-controlled glassy carbon electrode by Au-S bond, which was then hybridized with the target miRNA-16. In the presence of bases, primer, and polymerase, the polymerization began, leading to the release of miRNA-16. After the hybridization reaction between the probe DNA (PDNA) and the remaining sequence of the CP’s stem component and formation of the core–shell sun-like structure, they embedded pyridine ruthenium (Ru(bpy)_3_^2+^) complex into the assistance DNA (ADNA), which was loaded on a nanogold surface. Lastly, the composite of AuNPs@(PDNA+ADNA- Ru(bpy)_3_^2+^ was added and the ECL intensity was recorded. Due to the polymerization cycle and aggregation of Ru(bpy)_3_^2+^ as the illuminator, they were able to significantly increase the sensitivity and detect miRNA 16 at a very low concentration, with a detection limit of 4.3 × 10^−17^ cells/mL.

### 4.3. Electrical Biosensors

Electrical biosensors provide label-free, rapid, highly sensitive and direct detection of biomolecules. In electrical biosensors, capturing a target by a biological receptor results in a change in potential, impedance, or current that can be correlated with target concentration. Recently, electrical biosensors based on semiconductor nanowires and nanoribbons have been considered for real-time and label-free detection of macromolecules and cancer biomarkers at subfemtomolar concentrations [96,97]. Silicon nanoribbons as extended silicon nanostructures have been recently introduced as highly sensitive (bio)sensor chips for potential use in medicine and biotechnology when at least femtomolar or lower detection is required [98,99,100]. Ivanov et al. [101] developed a silicon nanoribbon-based DNA-biosensor for the detection of circular RNA nuclear factor IX (circNFIX) as a molecular biomarker of glioma in humans (Figure 6). Using the proposed assay, they were able to detect circNFIX in plasma samples from glioma patients with a limit of detection of 1.1 × 10^−17^ M in real time. A summary of developed biosensors for childhood cancer is shown in Table 1.

A list of the abbreviations used in this review is provided in Table 2.

## 5. Conclusions and Future Outlook

This review highlights the concern about childhood cancers and the importance of biomarker detection for early diagnosis and prognosis of malignancies in this age group. Biosensors are shown to be an accurate, non-invasive, and rapid technology for detecting specific biomarkers of childhood cancers. However, biosensors should have high sensitivity to detect very low levels of biomarkers at the early stages of cancer. The application of nanotechnology in developing biosensors has had a tremendous impact on the sensitive detection of biomarkers at the trace level, allowing earlier diagnosis of cancer and improved patient survival. On the other hand, detecting a panel of biomarkers associated with a particular cancer type may reduce diagnostic time and ultimately positively impact clinical outcomes. In this context, the integration of nanomaterials into electrochemical biosensors makes them the best candidates for future clinical applications in simultaneous measurement of multiple biomarkers, because of their ease of use, their fast and highly sensitive response, and the possibility of miniaturization as well as the development of point-of-care devices. Considering the limited research on biosensors for childhood cancers, studies and development of ultrasensitive and wearable biosensors for clinical use have great potential, not only for non-invasive and cost-effective diagnosis of pediatric cancers, but also for monitoring the patient’s treatment progress. However, a comprehensive understanding of molecular changes, cancer cell behavior, and associated biomarkers, as well as the mechanisms of interaction between nanomaterials on the surface of biosensors and various biomarkers, are the major challenges in biosensor design and require further investigation.

## Figures and Tables

**Figure 1 sensors-23-01482-f001:**
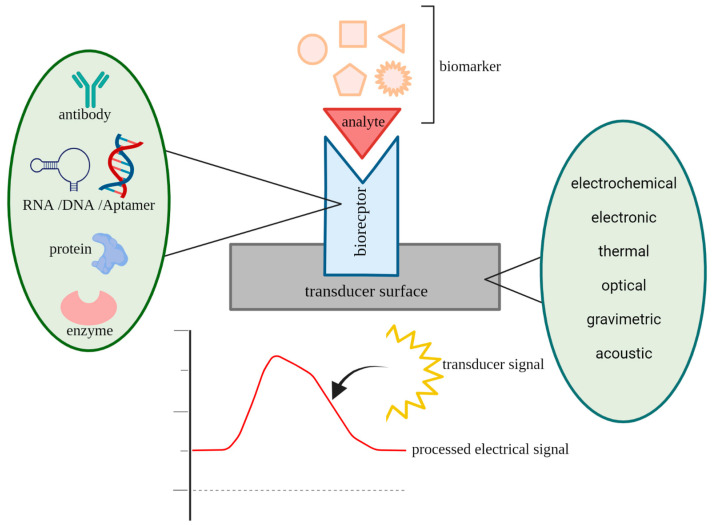
Schematic picture of a biosensor’s construction and function.

**Figure 2 sensors-23-01482-f002:**
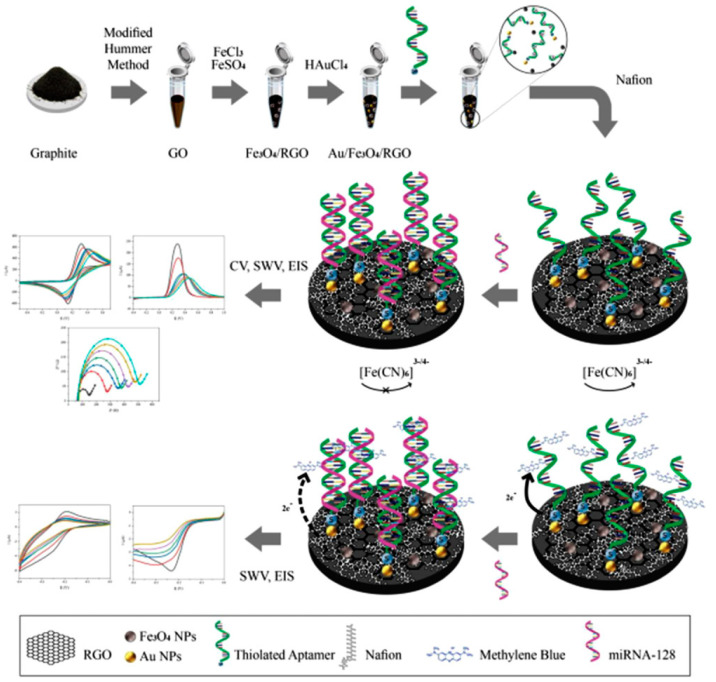
Schematic representation of Au NPs/Fe_3_O_4_/rGO nanocomposite preparation, immobilization of label-free as well as labeled aptamer on the surface of Au NPs/Fe_3_O_4_/rGO, and electrochemical analysis for determination of miRNA-128 concentration [23], reprinted with permission from WILEY.

**Figure 3 sensors-23-01482-f003:**
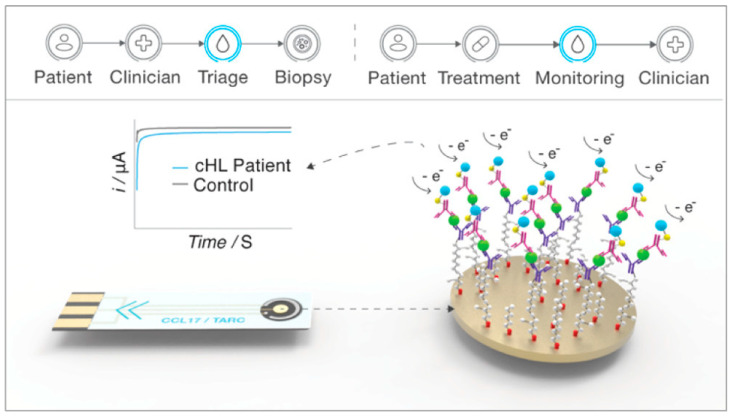
Schematic illustration of electrochemical immunosensor as a point-of-care test for detecting CCL17/TARC [72], reprinted with permission from American Chemical Society.

**Figure 4 sensors-23-01482-f004:**
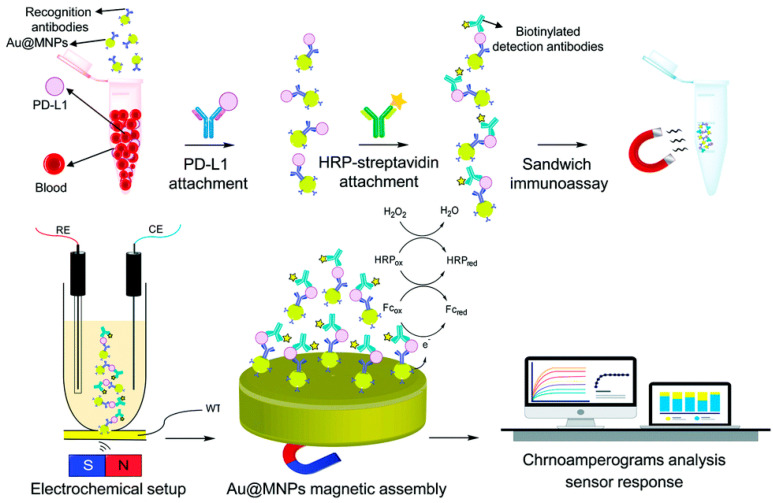
Schematic representation of working principle of the steps involved in the proposed sensing strategy for quantifying PD-L1 in undiluted whole-blood media [90], reprinted with permission from Royal Society of Chemistry.

**Figure 5 sensors-23-01482-f005:**
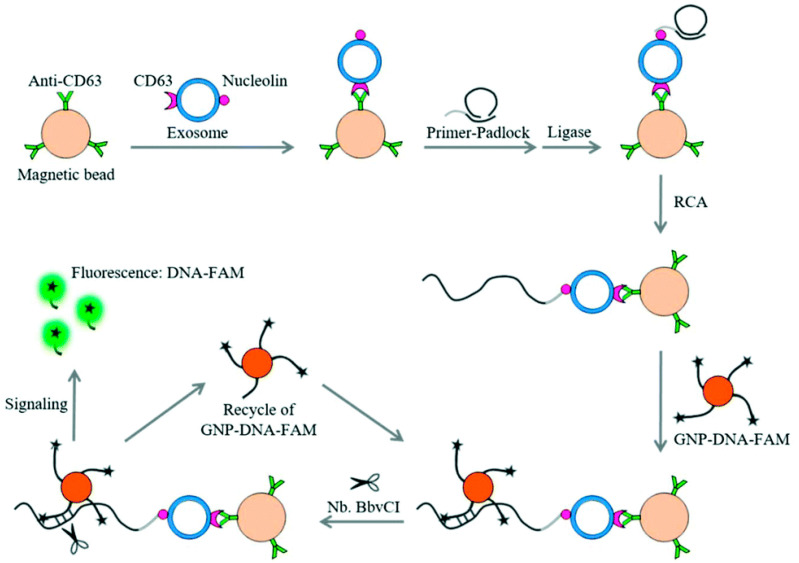
Schematic illustration of the dual-signal amplification-based platform for the ultrasensitive detection of exosomes. The primer contains the RCA primer (labeled in black) and AS1411 aptamer (labeled in gray) [91], reprinted with permission from Royal Society of Chemistry.

**Figure 6 sensors-23-01482-f006:**
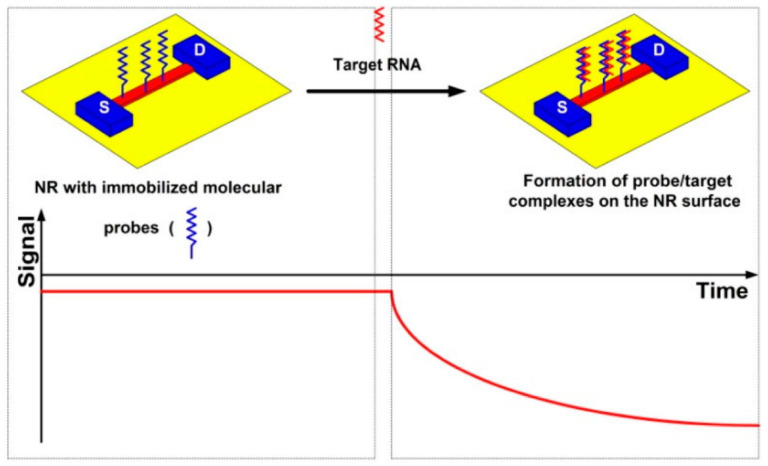
A schematic of the principle of the real-time nanoribbon (NR)-based RNA detection [101], reprinted with permission from MDPI.

**Table 1 sensors-23-01482-t001:** Summary of developed biosensors for childhood cancers.

Biosensor Material	Detection Method	Biomarker	Ref.
DNA-aptamer/COD ^1^-Au NPs	Electrochemical	BCR-ABL1/CEA	[88]
NA/Au-NPs- PANI ^2^	Electrochemical	BCR-ABL	[89]
Aptamer/Au NPs-Fe_3_O_4_-rGO ^3^	Electrochemical	miRNA-128	[23]
Aptamer/PEDOT ^4^-Au NPs	Electrochemical	CCRF-CEM cells	[24]
Aptamer/Au NPs-Fe_3_O_4_-rGO	Electrochemical	CCL17/TARC	[72]
Protein/Zr-MOFs ^5^ Antibody/labeled DNA/Au NPs Antibody/ExTFG-Au ^6^ Antibody/Ag@AuNIs ^7^ Aptamer/Fe_3_O_4_@TiO_2_ Aptamer/Fe_3_O_4_@Au NPs DNA/silicon	Electrochemical Electrochemical Optical Optical Optical Optical Optical Electrical	GBM-derived exosome PD-L1 leukemia-derived exosomes PD-L1 MCT4 ^8^ PD-L1 miRNA-16 circNFIX ^9^	[43] [90] [91] [92] [93] [94] [95] [101]

^1.^ COD: carbon quantum dots, ^2.^ PANI: polyaniline, ^3.^ rGO: reduced graphene oxide, ^4.^ PEDOT: poly(3,4-ethylenedioxythiophene), ^5.^ Zr-MOF: Zr/metal-organic frameworks, ^6.^ excessively tilted fiber grating, ^7.^ silver nanoparticles decorated on gold nano-islands, ^8.^ MCT4: monocarboxylate transporter 4, ^9.^ circNFIX: circular RNA nuclear factor IX.

**Table 2 sensors-23-01482-t002:** List of abbreviations used in this review.

Abbreviation	Explanation	Abbreviation	Explanation
LMICs	Low- and middle-income countries	AML1-ETO	Fusion protein resulting from t(8;21)(q22;q22) translocation
CNS	Central nervous system	PML-RARA	Promyelocytic leukemia/retinoic acid receptor alpha
CSF	Cerebrospinal fluid	PF4	Platelet factor 4
CT	Computed tomography	CTAP III	Connective tissue-activating peptide III
US	Ultrasound	C3a	Complement component 3a
MRI	Magnetic resonance imaging	miRNA	Micro ribonucleic acid
US	Ultrasonography	CCRF-CEM	Human leukemic lymphoblast
SPECT	Single-photon emission computed tomography	MB	Medulloblastoma
PET	Positron emission tomography	TMSB4X	Tripartite motif-containing protein 33
DNA	Deoxyribonucleic acid	HSPs	Heat shock proteins
RNA	Ribonucleic acid	GBM	Glioblastoma
WBCALL	White blood cell Acute lymphoblastic leukemia	ALK	Anaplastic lymphoma receptor tyrosine kinase gene
AML	Acute lymphoblastic leukemia	MIBG	Meta-iodobenzylguanidine
CML	Chronic myeloid leukemia	NB	Neuroblastoma
CD	Cluster of differentiation	FDG	Fluorodeoxyglucose
TDT	Terminal deoxyribonucleotidyl transferase	MYCN	N-myc proto-oncogene protein
T-ALL	Thymic ALL	PIM kinase	Pim-1 proto-oncogene, serine/threonine kinase
BCR-ABL	Chimeric gene of BCR and ABL	TARC	Thymus and activation-regulated chemokine
TEL-AML1	Fusion gene resulting from t(12;21) translocation	CCL17/TARC	C-C Motif Chemokine Ligand 17/thymus and activation-regulated chemokine
FUBP1	Far-Upstream Element-Binding Protein 1	cHL	Classic Hodgkin lymphoma
UCHL1	Ubiquitin C-Terminal Hydrolase L1	NF-κB	Nuclear Factor Kappa B
TMOD	Tropomodulin1	S100A8	S100 calcium-binding protein A8
NSE	Neuron-specific enolase	LRG1	Leucine-rich alpha-2-glycoprotein 1
CTCs	Circulating tumor cells	ELISA	Enzyme-linked immunosorbent assay
3-MT	3-Methoxytyramine		
HVA	Homovanillic acid	PCR	Polymerase chain reaction
VMA	Vanillylmandelic acid	CEA	Carcinoembryonic antigen
NMN	Normetanephrine	Au NPs	Gold nanoparticles
VLA	Vanillactic acid	Fe_3_O_4_	Iron oxide
WT	Wilms tumor		
WAGR	Wilms tumor, aniridia, genitourinary problems and range of developmental delays	rGO	Reduced graphene oxide
LOH	Loss of heterozygosity	H_2_O_2_	Hydrogen peroxide
PD-L1	Programmed death-ligand 1	Sulfo-LC-SPDP	Sulfosuccinimidyl 6-[3′-(2- pyridyldithio)propionamido]hexanoate
HL	Hodgkin lymphomas	Zr-MOFs	Zr/metal-organic frameworks
NHL	Non-Hodgkin lymphomas	EGFR	Epidural growth factor receptor
Au@MNPs Fc SPR LSPR CL ALS ECL circNFIX NR	Gold-coated magnetic nanoparticles Ferrocenemethanol Surface plasmon resonance Localized surface plasmon resonance Chemiluminescence Alkaline phosphates Electrochemiluminescence Circular RNA nuclear factor IX nanoribbon	HRP RCA FAM ExTFG Au NIs MCT4 EVs TiO_2_ Ru(bpy)_3_^2+^	Horseradish peroxidase Rolling circle amplification Fluorescent-label Excessively tilted fiber grating Au nano-islands Monocarboxylate transporter 4 Extracellular vehicles Titanium dioxide Pyridine ruthenium complex

## Data Availability

Not applicable.

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
