# Peer review of "Biomarkers and Corresponding Biosensors for Childhood Cancer Diagnostics"

_sensors, 2023, doi:10.3390/s23031482_

Round 1

Reviewer 1 Report

Authors reviewed biomarkers and corresponding biosensors for childhood cancer diagnostics. This is a new perspective, and I believe this article can provide useful information to colleagues who care about childhood cancer.

 The authors summarized all biomarkers for childhood cancer. Biosensor studies of childhood cancer biomarkers are also presented in the paper.

 Overall, this review can be published, only some details need to be revised.

 For example, “4.2. Electrical biosensors” should be “4.3 …”, “4. Conclusion and future outlook” should be “5. Conclusion and future outlook”

 It is best to double check the text and content of the article.

Author Response

Authors reviewed biomarkers and corresponding biosensors for childhood cancer diagnostics. This is a new perspective, and I believe this article can provide useful information to colleagues who care about childhood cancer.

 The authors summarized all biomarkers for childhood cancer. Biosensor studies of childhood cancer biomarkers are also presented in the paper.

 Overall, this review can be published, only some details need to be revised.

 For example, “4.2. Electrical biosensors” should be “4.3 …”, “4. Conclusion and future outlook” should be “5. Conclusion and future outlook”

 It is best to double check the text and content of the article.

We would like to thank you sincerely for the preliminary review of this manuscript. As suggested, we have checked the text and content of the article and revised it based on the reviewers' comments.

Reviewer 2 Report

The results are interesting and I recommend its publication with minor revision.

1.      What  type of sensor materials give more reliable results

2.      Sensor sensitivity table missing

3.      What about real sample analysis

4.      What are the difficulties and advantages

5.      Future perspectives missing   

6.      In the introduction why the authors have chosen this sensor for detection

7.      Electrochemical methods contains many methods, please mention specific technique

8.       Many other methods did analysis,  if possible please include

Author Response

The results are interesting and I recommend its publication with minor revision.

  1. What type of sensor materials give more reliable results
  2. Sensor sensitivity table missing
  3. What about real sample analysis
  4. What are the difficulties and advantages
  5. Future perspectives missing
  6. In the introduction why the authors have chosen this sensor for detection
  7. Electrochemical methods contains many methods, please mention specific technique
  8. Many other methods did analysis, if possible please include.

We would like to sincerely thank you. We tried to response the comments, but regarding to the following reasons we think there was a mistake and these comments do not refer to our manuscript. First, our manuscript is a review paper and we doesn’t have any results (comment 1, 2, 3). The comments are general without referring to the parts or lines, where we should consider them. Also, we have mentioned future perspectives in the conclusion and future outlook (comment 5). We did not choose and write about any sensor for detection in the introduction (comment 6). For each of cancer type different diagnostic methods are described (comment 8). For every section advantages and disadvantages are mentioned (comment 4).  

Reviewer 3 Report

1. A graphic summary should accompany the manuscript.

2. References must contain the abbreviated name of the journal, see the Reference List and Citations Guide for more detailed information.

3. Please explain to readers, what DNA, RNA, CD10, CD19, CD20, CD22, CD 24, CD34, TdT, PML-RARA, BCR-ABL, Fe3O4, H2O2, Sulfo-LC-SPDP, CCL17/TARC, PD-L1, HRP,

4. Table 1 should appear under the subsection Electrical biosensors.

5. Figure 1. Schematic image of the construction and operation of a biosensor. is it by the authors?

6. What period does the review cover?

Author Response

  1. A graphic summary should accompany the manuscript.

We would like to thank you for the thoughtful comments and efforts towards improving our manuscript. We provided a graphical abstract ad added to manuscript.

  1. References must contain the abbreviated name of the journal, see the Reference List and Citations Guide for more detailed information.

Thank you for your comment. We have used the Citavi program for inserting the references and in this program, we have chosen the Sensors MDPI as the journal for references. So, it has been done according to this software for Sensors.

  1. Please explain to readers, what DNA, RNA, CD10, CD19, CD20, CD22, CD 24, CD34, TdT, PML-RARA, BCR-ABL, Fe3O4, H2O2, Sulfo-LC-SPDP, CCL17/TARC, PD-L1, HRP à

Thank you for your comment. We added a table for the used abbreviations in this review.

  1. Table 1 should appear under the subsection Electrical biosensors.

We agree. It was replaced.

  1. Figure 1. Schematic image of the construction and operation of a biosensor. is it by the authors?

Yes. We designed it ourselves.

  1. What period does the review cover?

We have attempted to cover all work reported to date, and therefore this overview covers work over a period from 1995 to 2022.